# Image- and Fluorescence-Based Test Shows Oxidant-Dependent Damages in Red Blood Cells and Enables Screening of Potential Protective Molecules

**DOI:** 10.3390/ijms22084293

**Published:** 2021-04-20

**Authors:** Manon Bardyn, Jérôme Allard, David Crettaz, Benjamin Rappaz, Gerardo Turcatti, Jean-Daniel Tissot, Michel Prudent

**Affiliations:** 1Laboratoire de Recherche sur les Produits Sanguins, Transfusion Interrégionale CRS, CH-1066 Epalinges, Switzerland; manon.bardyn@itransfusion.ch (M.B.); jerome.allard@polymtl.ca (J.A.); david.crettaz@itransfusion.ch (D.C.); 2Département de Génie Chimique, École Polytechnique de Montréal, Montréal, QC H3C 3A7, Canada; 3Biomolecular Screening Facility (BSF), Ecole Polytechnique Fédérale de Lausanne (EPFL), CH-1015 Lausanne, Switzerland; benjamin.rappaz@epfl.ch (B.R.); gerardo.turcatti@epfl.ch (G.T.); 4Faculté de Biologie et de Médecine, Université de Lausanne, CH-1011 Lausanne, Switzerland; doyen.fbm@unil.ch; 5Center for Research and Innovation in Clinical Pharmaceutical Sciences, Institute of Pharmaceutical Sciences of Western Switzerland, University of Lausanne, CH-1011 Lausanne, Switzerland

**Keywords:** red blood cell, transfusion, antioxidant, oxidant, high-throughput screening, digital holographic microscopy, fluorescence

## Abstract

An increase of oxygen saturation within blood bags and metabolic dysregulation occur during storage of red blood cells (RBCs). It leads to the gradual exhaustion of RBC antioxidant protective system and, consequently, to a deleterious state of oxidative stress that plays a major role in the apparition of the so-called storage lesions. The present study describes the use of a test (called TSOX) based on fluorescence and label-free morphology readouts to simply and quickly evaluate the oxidant and antioxidant properties of various compounds in controlled conditions. Here, TSOX was applied to RBCs treated with four antioxidants (ascorbic acid, uric acid, trolox and resveratrol) and three oxidants (AAPH, diamide and H_2_O_2_) at different concentrations. Two complementary readouts were chosen: first, where ROS generation was quantified using DCFH-DA fluorescent probe, and second, based on digital holographic microscopy that measures morphology alterations. All oxidants produced an increase of fluorescence, whereas H_2_O_2_ did not visibly impact the RBC morphology. Significant protection was observed in three out of four of the added molecules. Of note, resveratrol induced diamond-shape “Tirocytes”. The assay design was selected to be flexible, as well as compatible with high-throughput screening. In future experiments, the TSOX will serve to screen chemical libraries and probe molecules that could be added to the additive solution for RBCs storage.

## 1. Introduction

Red blood cells (RBCs) work in a high O_2_ content environment and therefore possess a robust system to fight against oxidative stress. The imbalance between oxidants and antioxidants that generates this stress is counteracted by chemical-based antioxidant systems, such as lipophilic tocopherols, ubiquinol, carotenoids and flavonoids, hydrophilic ascorbic acid (AA), uric acid (UA) and thiols, as well as enzyme-based systems such as glutathione peroxidase or reductase, superoxide dismutase. Nevertheless, in the context of transfusion medicine, the storage of RBCs as RBC concentrates (RCCs) induces an oxidative stress that plays a major role in the apparition of the so-called storage lesions [1,2].

Previous studies have explored the opportunity to compensate the apparition of oxidative stress by adding antioxidants (or precursors of them) in the preservative solution. The addition of AA, UA, precursors of glutathione or components such as vitamin E provided limited improvements of the storage lesions, whatever the parameters followed [1,3,4,5,6]. Investigators evaluated supplementation with ascorbate and urate (with the aim to restore normal plasma concentrations) in order to prevent the loss of antioxidant power (AOP) and metabolic dysregulation subsequent to blood products preparation [7]. Despite the observed variations in metabolism, this treatment did not improve the “standard aging” parameters (i.e., such as morphology changes and hemolysis) [3]. Such mitigated results triggered the following reflections: (1) Are these compounds indeed protective against oxidative stress? (2) Are they efficient in such context, i.e., this particular environment, cell type, time point, etc.? (3) Are the tested concentrations high enough? (4) Are the readouts sensitive or specific enough?

Different strategies are generally used to evaluate the improvement brought by a treatment. Some are very basic such as hematological parameters, hemolysis rate and measurement of targeted metabolites; others adopt advanced technologies aiming to decipher multiples biochemical dimensions by using omics. New approaches like microfluidic devices could be used to evaluate the impact of aging or chemical treatments on RBC deformability [8,9]. Finally, cell morphology using imaging flow cytometry or digital holographic microscopy (DHM) are useful tools to provide information of RBC phenotype [10,11,12,13].

Antioxidants can be measured by chemical-based assay, such as the well-known oxygen radical absorbance capacity (ORAC) assay that only observes the direct quenching capacity of an antioxidant toward an oxidant, the total antioxidant capacity (TAC), or by using electrochemical approaches [7,14,15]. Oxidative stress can be monitored using redox-sensitive probes such as the 2′-7′-dichlorofluorescin diacetate (DCFH-DA) molecule, as shown in 1999 by Wang and Joseph [16]. In 2007, Wolfe and Liu developed a cellular antioxidant assay (CAA) on HepG2 cells with the aim to test the antioxidant activity of phytochemical food extracts and dietary supplements [17]. Honzel et al. proposed a cell-based antioxidant protection in an erythrocyte model (CAP-e) in 2008 [18].

Here, we tested a set of antioxidant molecules at various concentrations to see if they were protective against different kind of oxidants. To do so, we used a test of sensitivity against oxidative stress (TSOX), of which outputs are based on reactive oxygen species (ROS) detection by fluorescence and on observation of cell morphology by DHM.

## 2. Results

### 2.1. Workflow of the Test of Sensitivity to Oxidative Stress (TSOX)

In the present study, the workflow consisted of dispensing RBCs in multiwell plates (96 wells presently), in order to test several conditions in parallel (Figure 1A).

First, the RBCs were pre-incubated for one hour with AA, UA, trolox or resveratrol antioxidants. Any kind of molecules could be tested in theory. Then, oxidants (i.e., 2,2′-Azobis(2-methylpropionamidine) dihydrochloride [AAPH], diamide and hydrogen peroxide [H_2_O_2_]) were added (Figure 1B). The selected oxidants have different mode of action. The first one, AAPH, is a water-soluble azo compound that generates peroxyl radicals (ROO^•^) and exhibits temperature-dependent degradation at constant rate [19]. The second, diamide, is a thiol-oxidizing agent that oxidizes sulfhydryl groups to disulfide form. The third one, H_2_O_2_, is a water-soluble oxidizing agent and inorganic peroxide.

The RBCs treated with both protective and stress-generating molecules were followed overnight using either a readout based on fluorescence emission or one on morphology. Of note, higher oxidant concentrations were used for morphology analysis, because label-free DHM was less sensitive. This was probably due to it giving information about the change of phenotype triggered by the oxidative effect of a compound, whereas fluorescence emission by DCFH-DA is a direct indicator of the quantity of ROS produced. The concentrations were therefore adapted to the readouts.

For fluorescence analysis, the DCFH-DA molecule was selected [20]. This reporter probe is activated when intracellular ROS are generated. A microplate fluorometer was used for signal quantification in each well.

The second readout proposed is based on morphology analysis by DHM, of which outputs are phase images encoding a quantitative value (i.e., optical path difference [OPD]) measured on each pixel of the phase image [12,21]. OPD value at the (x,y) position is proportional to the cell thickness $d_{\left(x,y\right)}$ and the difference of refractive index between the cells ($n_{cell\left(x,y\right)}$) and surrounding medium ($n_{m}$) (Equation (1)). RBCs refractive index n, which reflects cellular internal composition, is mostly defined by the hemoglobin concentration, and their thickness d is strongly impacted by their morphology [21].
(1)$$OPD_{\left(x,y\right)}=d_{\left(x,y\right)}\times \left(n_{cell\left(x,y\right)}-n_{m}\right)$$

For the TSOX, the average of the OPD distribution (OPD AVG) and confluency parameters were considered. The confluency refers to the percentage of the phase image occupied by RBCs. The evolution of this latter parameter is impacted by cell shape change, swelling or shrinkage and is strongly reduced in case of hemolysis. In general, confluency was normalized (divided by the confluency measured right after oxidant addition), as this parameter depends on the cell seeding.

### 2.2. Effect of Oxidant Treatment on the Red Blood Cells

The effect of the three selected oxidants alone (i.e., 0 µM antioxidants condition) was evaluated with both readouts (Figure 2).

Fluorescence emission over time following RBC treatment with 250 µM or 1 mM AAPH, with 250 µM or 1 mM diamide, or with 0.0001 or 0.001% H_2_O_2_ was evaluated. From time-lapse analysis, it appeared that the slope (an indicator of ROS generation rate) was steeper with increasing oxidant concentrations (Figure 2A). Similarly, the calculated area under the curve (AUC), informing about the global quantity of ROS produced, increased following treatment, and all conditions, except for 0.0001% H_2_O_2_, were statistically different from the “no-oxidant” control (Figure 2B). Knowing that the RBCs were treated with a similar amount of reporter probe, the fact that some conditions reached lower maximal plateau values could indicate either that all oxidant molecules were used, quenched or degraded. It is interesting to notice that fluorescence slightly increased in the control condition where no oxidant was added in the well. It suggests that the incubation conditions triggered spontaneous generation of endogenous and/or exogenous ROS.

Then, the effect of the same three oxidants on RBC morphology was assessed by DHM. Of note, higher oxidant concentrations were used (i.e., 2 or 5 mM AAPH, 1 or 2 mM diamide, or 0.001 or 0.005% H_2_O_2_). Time-lapse analysis of OPD AVG and confluency is presented in Figure 2C and the corresponding AUC in Figure 2D. The modifications of cell morphology were dependent on the type of oxidant as clearly depicted in Figure 2E. With AAPH and diamide, RBC shape was maintained for a short lag time. In diamide-treated samples, the sudden increase of OPD AVG corresponded to the transformation of discocytes into echinocytes exhibiting large spicules and loosing large portions of membrane, and then spherocytes (maximum of the curve). Spherocytes generate higher OPD signal because they are thicker and denser. Both OPD AVG and confluency parameters then dropped when cells started to lyse as shown in Figure 2C. AAPH treatment also ultimately led to cell lysis. Despite the fact that H_2_O_2_ increased ROS generation within RBCs, its effect on morphology and more particularly on confluency was limited, even with higher concentrations (i.e., 0.01 and 0.1% H_2_O_2_; data not shown).

### 2.3. Treatment of the Red Blood Cells with Antioxidants

The TSOX was then used to determine if the addition of 10, 100 or 1000 µM of AA, UA, trolox and resveratrol demonstrated protective effect.

The use of 96-well multiwell plates enable to test up to 96 conditions in parallel. For this assay, 48 combinations were tested in duplicate on each plate, with one plate per oxidant and per RCC unit (Figure 1B). To have a global vision of the results of our “miniscreen” (all data are available as Appendix A), we choose to represent the mean AUC calculated over time for signals obtained with fluorescence and morphology (OPD AVG and confluency) readouts, using a colored heat map (Figure 3). Comparison of the AUC between the 10, 100 and 1000 µM antioxidant concentration and the “no-antioxidant” condition (0 µM, which corresponds to the left cells of each box) was tested by two-way ANOVA. Significant differences are indicated by stars.

Some antioxidants produced an effect even in absence of any oxidants. Hence, AA and trolox (and resveratrol only in one case) treatment lowered basal fluorescence. On the contrary, it seems that UA rather favored ROS generation. Resveratrol at 1000 µM significantly increased the AUC calculated for the OPD AVG parameter (a detailed analysis of the phenomenon is provided in the next section).

Antioxidant addition efficiently limited fluorescence emission triggered by AAPH, diamide and H_2_O_2_, except for UA. Indeed, UA was not able to counteract the AAPH-induced ROS at these concentrations, and the effects were lower compared to other antioxidants for diamide- and H_2_O_2_-induced stress. Moreover, the protection depended on the concentration of antioxidant. Looking at the OPD AVG and confluency parameters, it appeared that AA was less protective than trolox and resveratrol at similar concentrations.

The best coherence between fluorescence and morphology results was obtained when RBCs were treated with diamide. On the contrary, significant discrepancies were observed between the two methods with H_2_O_2_.

### 2.4. Detailed Analysis: Effect of Resveratrol on Red Blood Cells

As a detailed example, the results obtained with resveratrol are presented in Figure 4. This compound was chosen as its capacity to protect RBCs against AAPH, and diamide was visible with both readouts. However, and as seen before, the changes triggered by H_2_O_2_ were not captured with the OPD AVG and confluency parameters (non-significant difference between 0 µM resveratrol and control samples, in which RBCs have neither been treated with an oxidant nor an antioxidant).

For RBCs treated with AAPH and diamide, fluorescence emission was reduced proportionally to the amount of antioxidant added (Figure 4A,B). However, the effect was not total (compared to the control) even with 1000 µM resveratrol. The protection provided by resveratrol against those two oxidants was confirmed by the DHM analysis (Figure 4C,D). Here, treatment with 10 µM resveratrol did not provide protection against the changes of morphology triggered by 5 mM AAPH and 2 mM diamide treatment. Effects of resveratrol were significant at 100 and 1000 µM. Of note, the OPD AVG plateau value was increased when RBCs were exposed to 1000 µM resveratrol.

We observed an effect of resveratrol antioxidant on RBCs even in absence of oxidant molecules (Figure 5). Figure 5A shows the effect of 0, 10, 100 and 1000 µM resveratrol on fluorescence emission and morphology (OPD AVG and confluency). All three concentrations significantly lowered the fluorescence in comparison to the no-antioxidant/no-oxidant control (0 µM condition).

Immediately after the addition of 1000 µM resveratrol, the OPD AVG increased. During incubation, most of the RBCs first transformed into stomatocytes, which then became smaller and finally lysed, leading to a gradual reduction of the confluency. Interestingly, some of the stomatocytes recovered back to normal shape and then transformed into diamond-shaped “Tirocytes” (Figure 5B and red circles in Figure 5C).

## 3. Discussion

In this study, different oxidants (i.e., AAPH, diamide and H_2_O_2_) were tested. All three of them induced quantitative changes using fluorescence as an output, whereas only AAPH and diamide yielded visible morphological modifications.

The reaction kinetics of each oxidant was in accordance with their individual mechanism of action. Hence, AAPH, an hydrophilic azo compound degrading at a constant rate into peroxyl radicals, triggered an increase of the fluorescence signal, changes of morphology and ultimately hemolysis [19,22]. Diamide, a thiol oxidant targeting the sulfhydryl groups of glutathione and amino acids, induced sudden changes at the level of the RBC membranes with loss of large vesicles. It could be due to the mode of action of diamide that triggers crosslinking in proteins (through disulfide bridges formation) such as the spectrins, which are major components of RBC membranes [23]. Finally, the highly reactive H_2_O_2_, which is also an endogenous ROS, almost immediately produced an increase of fluorescence and OPD AVG. This hydrophilic molecule is able to cross cell membranes and contributes to the formation of highly reactive radicals. The hydroxyl generated by H_2_O_2_ has very short half-life, which explains the kinetics observed in Figure 2A, compared to peroxide radicals from AAPH and O_2_ that have longer half-life of a few seconds.

After testing the oxidants, marked differences of potent antioxidants were detected with the TSOX. UA, a well-known antioxidant in plasma, appeared to be less effective than other antioxidants against ROS (with modulations in function of the type of oxidations applied). It is indeed known that urate possesses antioxidant and pro-oxidant properties in hydrophilic and lipid environments, respectively [24,25]. It is for this reason that UA has been coupled to AA to prevent pro-oxidant effects [3,26]. Equivalent benefits on quenching ROS were reported with AA, trolox and resveratrol (fluorescence data). In spite of the lower antioxidant power of AA compared to polyphenols such as resveratrol (as shown by a lower median effective dose (ED50) for resveratrol compared to AA in ABTS assay), AA can compensate its antioxidant power by a faster kinetic and shorter time to scavenge radicals compared to resveratrol [27]. This phenomenon can therefore explain the observation made here. As for the effects on morphology, AA was less efficient than trolox and resveratrol against the induced stress. These results are in agreement with recently published data on RBC damages induced by AAPH. Using BODIPY as a probe of lipid peroxidation, half maximal inhibitory concentrations (IC50s) of 138, 87 and 36 μM were calculated for AA, trolox and resveratrol, respectively, showing the better potential of resveratrol on preserving membrane integrity [28]. The beneficial effect of this red-wine polyphenol has been previously reported. Tedesco et al. showed a reduced H_2_O_2_-induced RBC hemolysis when pre-incubated with wine extracts (particularly with the one containing a large amount of polyphenols), as well as lower levels of ROS, met-hemoglobin and malondialdehyde [29]. Similar data were obtained with hypoxanthine-xanthine oxidase experiments to in vitro-modeled RBC damages, where lower H_2_O_2_ concentration and reduced met-hemoglobin saturation were quantified in the presence of red wine, quercitine or resveratrol [30]. The effect of the latter was also investigated on RBC morphology. Despite the absence of total protection, resveratrol (at 80 μM) decreased the proportion of echinocytes and spheroechinocytes in their stress model.

Of particular interest, the higher concentrations of resveratrol (1 mM) induced what we called “tirocytes”. These diamond-shape RBCs require more investigations; however, a few published data might explain their formation. Polymerization of hemoglobin could contribute to the change in morphology as observed in sickle cell disease (holly leaf) [31]. Moreover and because of its poor water solubility, resveratrol interacts with membrane lipids. Depending on the membrane composition, it either inserts into the membrane or adsorbs onto the membrane (as shown by electrophysiological measurements on planar lipid membranes) [32]. The adsorption was observed in presence dioleoyl-phosphatidylserine and induced a curvature of the membrane. Defects in RBC membrane organization during the incubation (and aging) and adsorption of resveratrol could explain such diamond morphology. Shape changes in presence of resveratrol have also been observed with other cell lines. In K562 cells, it induced actin reorganization and spreading over a fibronectin matrix. This F-actin network formation was concomitant to tensin gene expression and protein production triggered by resveratrol (not occurring in anucleated RBCs), but a direct effect of this polyphenol could not be excluded [33]. Additionally, an atomic force microscopy study on chondrocytes treated by resveratrol has also shown increases of both F-actins and α-tubulin [34]. Elevated intracellular concentrations of resveratrol may also result in the formation of rigid crystal-like structures.

The approach chosen here made possible the evaluation of different molecules against cellular stress while being suitable for high-throughput screening (HTS) campaigns. Nevertheless, it has to be noticed that the cells were not washed between the antioxidant and the oxidant treatments. Indeed, exchange of the extracellular medium with nonadherent cells such as RBCs would have required to centrifuge the plate in order to sediment cells in suspension. Therefore, in the absence of washing, it is not possible to discriminate if the mode of action of the antioxidant was direct (e.g., ROS scavenger) and/or indirect (e.g., which sustains the RBC metabolism and thus promotes the recycling or the de novo synthesis of protective molecules).

Of course, the assay design presented here can be adapted if necessary. For example, the RBCs could be treated with the oxidant first and then the compounds of interest in order to evaluate their regenerative properties. A sensitization step, such as exposure to elevated temperature, high oxygen atmosphere, UV illumination or gamma irradiation, etc., could also be performed upstream to exacerbate the response of the RBCs to the treatments or the mimic cell preparation in transfusion medicine. One could also imagine treating the cells with a combination of molecules (as mentioned above for instance) having distinct properties in order to mimic more complex environments. Oxidants with other mode of action such as the liposoluble cumene hydroperoxide (CumOOH) that targets preferentially the inner part of the membrane lipid bilayer and causes lipid peroxidation, or the 2,2′-azobis(2,4-dimethylvaleronitrile) (AMVN) lipophilic azo compound that generates at constant rate radical formation within the lipid environment could also be used. Similarly, other reporter probes such as the BODIPY that sense lipid peroxidation can be used [35].

The curves obtained during time-lapse analyses are full of information about the reactions kinetics and could be used to decipher the mechanisms of action of the tested molecules. However, as discussed before, when a large number of conditions are analyzed in parallel, this kind of representation is too dense. With the aim to using the TSOX for HTS purposes, the AUC was calculated, as previously described, and a simplified view of the results using colored heat maps was proposed [18]. Other parameters could of course be extracted, such as the curve slopes, the lag-time, or the endpoints, and so forth.

Fluorescence and microscopy readouts selected provide complementary information. Hence, ROS quantification by fluorescence provides a direct observation of the effect of the oxidant/antioxidant molecules added. However, when used with a fluorometer in plates, it is relatively blind, as it only gives a bulk picture of what is happening within the whole well. This method of analysis is also sensitive to false-positive. Indeed, cell lysis reduces fluorescence, as it was the case for 1000 µM trolox treatment. A combination with DHM could thus be used to determine if the molecules are indeed true hits. DHM for which automation is available has already been validated for HTS applications [36,37]. Moreover, it can be mounted with a fluorescent camera.

Before using our test for screening libraries of chemical compounds, the next step is to validate statistically the TSOX by calculating for example the Z’-factor, a parameter widely used to guarantee the reliability, robustness and significance of an assay [38].

## 4. Materials and Methods

The blood used in the framework of this project was obtained from donors who gave their informed consent for the use of their blood components in research. The project was accepted by the Institutional Review Board of Transfusion Interrégionale CRS and agrees with local legislation.

Blood was processed according to Swiss requirements. Whole blood was collected in citrate-phosphate-dextrose (CPD), and the resultant RCCs were leuko-reduced and stored in saline-adenine-glucose-mannitol (SAGM) at 4 °C [39]. The mean storage duration for the RCCs used in this study was of 5.7 ± 2.4 days, with the youngest having been stored for 2 days and the oldest 11 days.

### 4.1. Antioxidant Protection and Oxidative Stress Induction

All conditions were tested in duplicates on the multiwell plate and repeated using three RCCs for Fluoroskan and three others for DHM experiments. RBCs were pre-incubated during 60 min with 0, 10, 100 or 1000 µM of antioxidant (i.e., AA, UA, trolox or resveratrol) before oxidant (i.e., AAPH, diamide or H_2_O_2_) addition for time-lapse analysis. For fluorescence analysis, the RBCs were treated with 0, 250 µM or 1 mM AAPH; 0, 250 µM or 1 mM diamide; and 0, 0.0001 or 0.001% H_2_O_2_. For DHM analysis, RBCs were exposed to 0, 2 or 5 mM AAPH; 0, 1 or 2 mM diamide; and 0, 0.001 or 0.005% H_2_O_2_.

### 4.2. Analysis of Reactive Oxygen Species Generation by Fluorometry

The DCFH-DA probe was used for ROS quantification within RBCs [40,41,42,43,44,45]. This molecule is nonionic and nonpolar, which enables its passage through plasma membranes. After entering the cells, the DA moiety is hydrolyzed (deacetylation) by the cellular esterases. Because the resulting DCFH molecule is polar, it is retained within the cell and is then transformed in its fluorescent counterpart (DCF) in presence of ROS. DCF is excited at a wavelength of 504 nm and emits at 529 nm (in ethanol), making it compatible with the common Fluorescein (FITC) filter set. For the analyses, DCFH-DA (Sigma-Aldrich, Steinheim, Germany) was prepared at 10 mM in 100% DMSO and aliquots were stored at −28 °C.

#### 4.2.1. Treatment of the Red Blood Cells with DCFH-DA Reporter Probe

RBCs sampled in CPD-SAGM RCCs (using a sterile syringe through a sample site) were washed two times in 0.9% NaCl (10 min of centrifugation at 2000× *g* and 4 °C). Then, the RBCs were resuspended at 10% hematocrit in 0.9% NaCl, treated with 50 µM DCFH-DA (0.5% DMSO) and incubated during 30 min at 37 °C under agitation to enable incorporation of the dye within cells. Finally, the tubes were centrifuged (as before), and the supernatant containing excess of DCFH-DA discarded.

#### 4.2.2. Quantification of Fluorescence Emission Using a Microplate Fluorometer

For fluorescence analysis, the RBCs stained with the DCFH-DA probe were diluted at 1% hematocrit in 0.9% NaCl, and 180 µL was dispensed in 96-well imaging plates. Then, 10 µL of antioxidant and, 60 min later, 10 µL of oxidant stock solutions (20×) were added for a final volume of 200 µL. The plate was incubated at 37 °C during the time-lapse analysis. The microplate fluorometer that was used is a Fluoroskan Ascent (and with the Ascent Software, Thermo Fisher Scientific, Shangai, China), equipped with a fluorometric filter pair (Ex/Em 492/527 nm).

Fluorescence was recorded seven times every 10 min during incubation with the antioxidant molecules alone (one hour in total) and then 60 times with 10 min interval after addition of oxidants. The whole analysis took 660 min.

### 4.3. Analysis of Morphological Changes Using Digital Holographic Microscopy

The impact of oxidative stress on RBC morphology was assessed using a DHM^®^ T1000 microscope (Lyncée Tec SA, Lausanne, Switzerland) equipped with a motorized microscope stage (Märzhäuser Wetzlar GmbH & CO. KG, Wetzlar, Germany), an incubator system (LCI Live Cell Instrument, Seoul, South Korea), and a 20×/0.40 NA objective (Leica Microsystems GmbH, Wetzlar, Germany), as previously described [12].

As before, RBCs were taken in CPD-SAGM RCCs and washed two times in 0.9% NaCl. The cell pellet was then resuspended in HEPA buffer and 80′000 RBCs (80 µL) were seeded per well in a 96-well black imaging plate coated with poly-L-ornithine. HEPA buffer is composed of 130 mM NaCl, 5.4 mM KCl, 1 mM CaCl_2_*2H_2_O, 0.5 mM MgCl_2_*6H_2_O, 10 mM glucose, 15 mM Hepes and 1 mg/mL BSA. To sediment the cells more rapidly, the plate was gently centrifuged (140× *g* for 2 min at room temperature). Then, 10 µL of antioxidant and, 60 min later, 10 µL of oxidant stock solutions (10×), were added per well (100 µL in total).

For imaging, the plate was put under the microscope in the incubation chamber set at 37 °C and 5% CO_2_. Four images were taken per well at 20× magnification. Four baseline images of the RBCs were first acquired before any treatment at 5 min interval. After addition of the antioxidants, the plate was imaged six times at 10 min intervals (60 min incubation). Twenty time-lapse images were finally taken once every hour after treatment with oxidants. The follow-up lasted a total of 1275 min (21 h15).

### 4.4. Data Analysis and Presentation

Prism (GraphPad Software Inc., San Diego, CA, USA) version 9.0.1 was used for data presentation and statistical analyses. The AUC was calculated (by the trapezoidal method) for time-lapse analyses, with a baseline set at 0 A.U. for fluorescence and 0 for confluency, and at 50 nm for OPD AVG (which corresponds to the average signal without RBC). Ordinary one- or two-way ANOVA with Dunnett’s multiple comparisons test was performed to compare the effect of the different antioxidant concentrations versus the no-antioxidant (0 µM) control.

## 5. Conclusions

The present work investigates the behavior of RBCs under various protective and stress treatments (oxidative stress here) and highlights different cell responses in function of the oxidant and antioxidant properties and their concentrations. Fluorescence readouts provides a direct and early quantification of the ROS production, with the need of a specific marker, whereas DHM provides secondary morphological alteration. The combination of both methods yields complementary data on the aging of RBCs in response to stress with the advantage of detecting false-positive hits.

The TSOX is intended to perform high-throughput analysis of large number of molecules, in order to potentially discover novel protective molecules. It is a valuable tool for investigating both the aging of RBCs under various conditions and discovering hit molecules that could be added in blood bags to help better preserve RBCs for transfusion. This tool also enables one to further investigate the relation between metabolic pathways and cell morphology. Moreover, the combination of assays with different designs could help understand the mode of action of toxic or protective compounds.

## Figures and Tables

**Figure 1 ijms-22-04293-f001:**
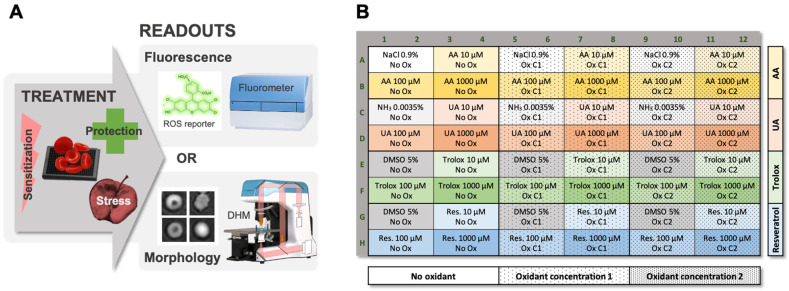
Principle of the TSOX Assay. (**A**) Red blood cells (RBCs) seeded in a microplate are treated with harmful (i.e., oxidant) and/or potentially protective molecules. Two possible readouts giving complementary information are proposed. The first one is based on the detection of fluorescence emitted by the DCFH-DA probe, when exposed to oxidative stress [20]. The second consists in the analysis by digital holographic microscopy (DHM) of the morphological changes triggered by the treatment(s). (**B**) Plate map (96 wells) used in the present study.

**Figure 2 ijms-22-04293-f002:**
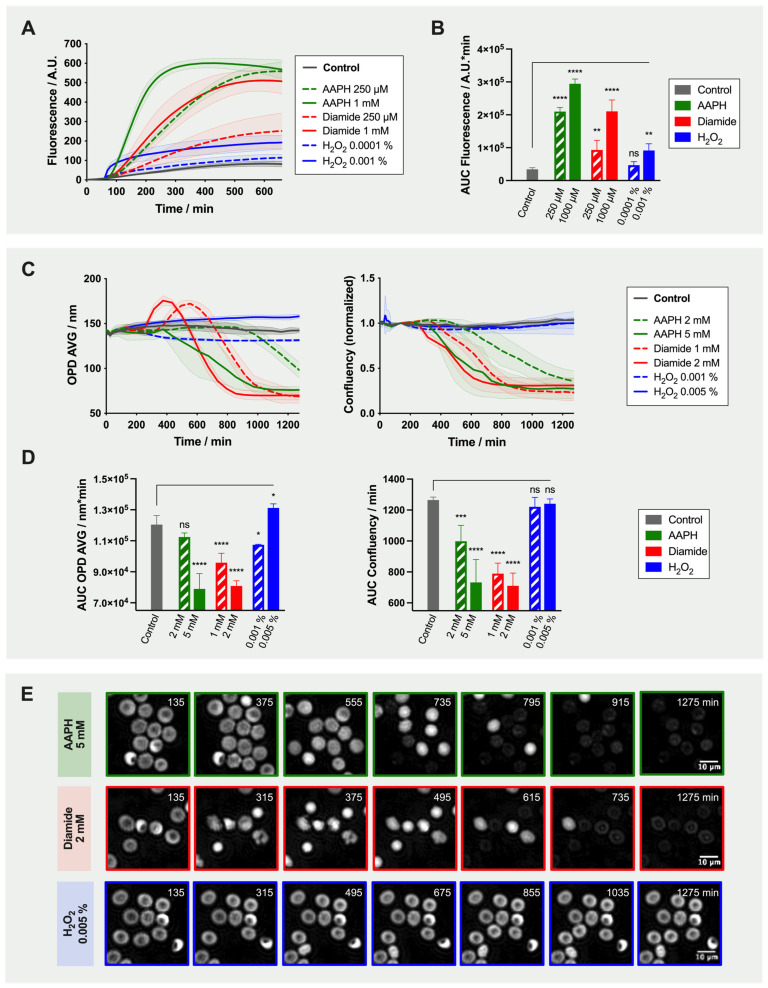
Time-Lapse Analysis of the Effects of Different Oxidants (i.e., AAPH, Diamide and H_2_O_2_) on the Oxidative Stress and Morphology of Red Blood Cells (RBCs). (**A**) Fluorescence emission induced by AAPH, diamide, H_2_O_2_ or no-oxidant treatments. (**B**) Area under the curve (AUC) calculated from fluorescence curves. (**C**) Morphological changes induced by AAPH, diamide, H_2_O_2_ or no-oxidant treatments. Left: average of the optical path difference distribution (OPD AVG) parameter; right: normalized confluency. (**D**) AUC calculated from OPD AVG and confluency curves. For the AUC, two-way ANOVA was performed to compare the effect of the three oxidants at different concentrations with a “no-oxidant” control; * *p*-value < 0.05, ** *p*-value < 0.01, *** *p*-value < 0.001, **** *p*-value < 0.0001, and “ns” non-significant. (**E**) Evolution of the morphology of RBCs treated with oxidants. Illustrative phase images acquired by digital holographic microscopy (DHM) at seven time points.

**Figure 3 ijms-22-04293-f003:**
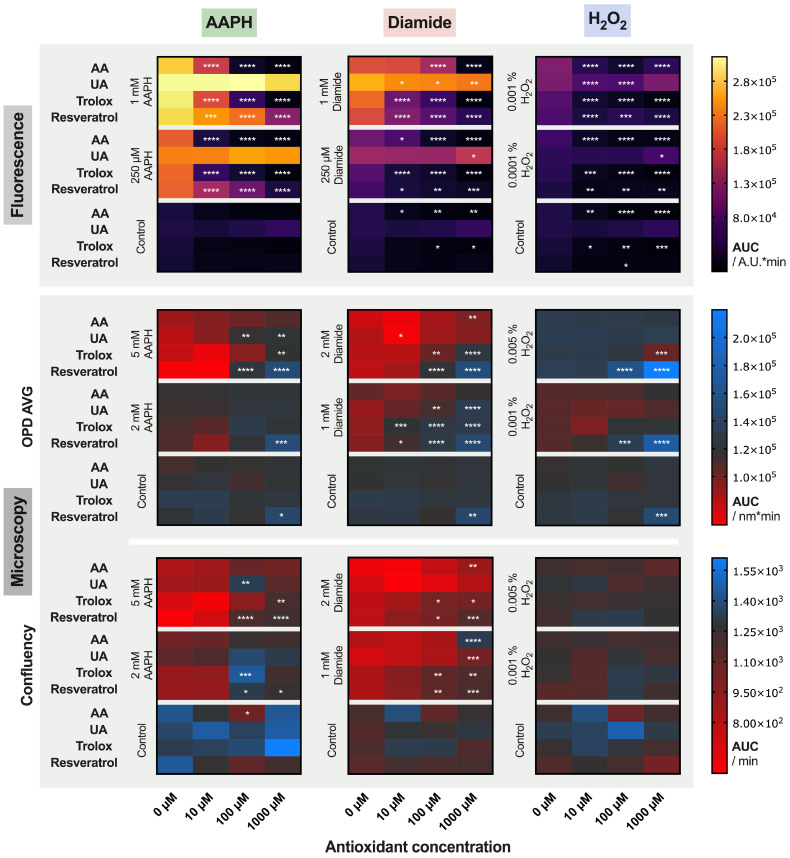
Summary Panel of the Effects on Red Blood Cells (RBCs) of 0, 10, 100 and 1000 µM Ascorbic Acid (AA), Uric Acid (UA), Trolox and Resveratrol against AAPH, Diamide and H_2_O_2_ at Different Concentrations. Top: effect on ROS by fluorescence emission; and bottom: effect on RBC morphology (i.e., average of the optical path difference distribution [OPD AVG] parameter and normalized confluency). Each cell on the heat map represents the mean area under the curve (AUC) for three RBC concentrates (RCCs) under a particular treatment. Two-way ANOVA was performed to compare the effect of the 0 µM versus 10, 100 or 1000 µM antioxidant conditions; * *p*-value < 0.05, ** *p*-value < 0.01, *** *p*-value < 0.001, and **** *p*-value < 0.0001.

**Figure 4 ijms-22-04293-f004:**
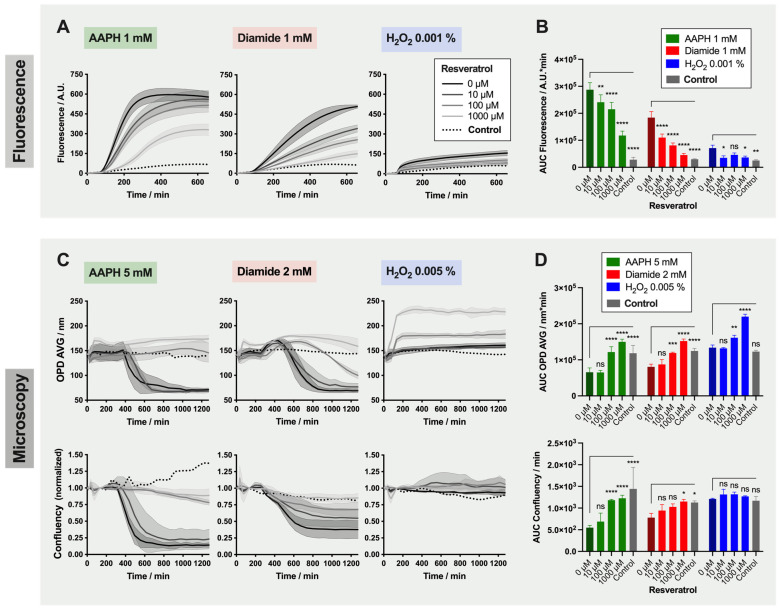
Assessment of the Protection Provided by Resveratrol against AAPH, Diamide and H_2_O_2_ Oxidants. (**A**) Time-lapse analysis of fluorescence emission for red blood cells (RBCs) treated with 1 mM AAPH, 1 mM diamide or 0.001% H_2_O_2_, and 0, 10, 100 and 1000 µM resveratrol. (**C**) Time-lapse analysis by digital holographic microscopy (DHM) of morphological changes triggered by 5 mM AAPH, 2 mM diamide or 0.005% H_2_O_2_, and 0, 10, 100 and 1000 µM resveratrol. top: average of the optical path difference distribution (OPD AVG) parameter; bottom: normalized confluency (**B**) and (**D**) area under the curve (AUC) calculated from fluorescence and OPD AVG and confluency curves. Of note: control samples were neither treated with oxidants nor antioxidants. “No-antioxidant” control corresponding to the 0 µM antioxidant samples are in darker color. For the AUC, two-way ANOVA was performed to compare the effect of resveratrol at different concentrations with the no-antioxidant condition; * *p*-value < 0.05, ** *p*-value < 0.01, *** *p*-value < 0.001, **** *p*-value < 0.0001, and “ns” non-significant.

**Figure 5 ijms-22-04293-f005:**
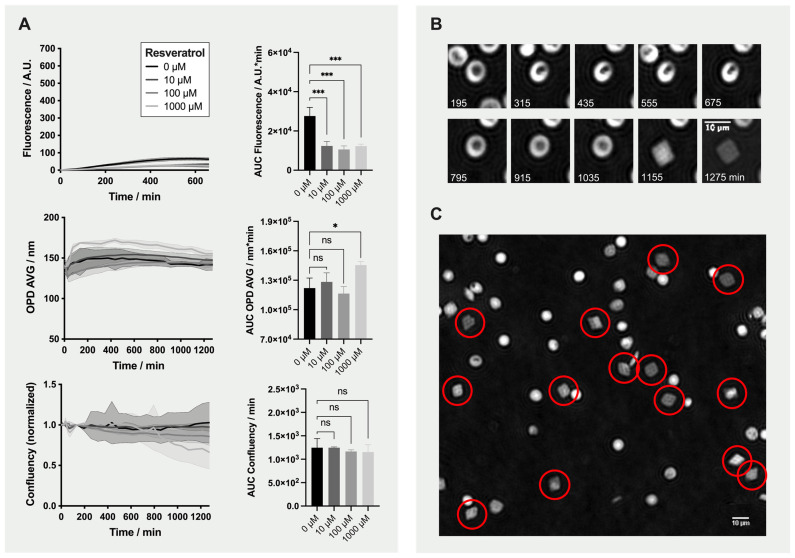
Effect of Resveratrol on Red Blood Cells (RBCs). Formation of “Tirocytes”. (**A**) Time-lapse analysis and the calculated AUC of fluorescence emission and morphology (i.e., average of the optical path difference distribution [OPD AVG] parameter and normalized confluency) for RBCs treated with 0, 10, 100 and 1000 µM resveratrol only (no oxidants). * *p*-value < 0.05, *** *p*-value < 0.001, and “ns” non-significant. (**B**) Illustrative phase images showing the morphological changes leading to a diamond-shaped RBC (treated with 1000 µM resveratrol). (**C**) Morphology of RBCs after 13h50 incubation with 1000 µM resveratrol. Diamond-shaped “Tirocytes” are surrounded by a red circle.

## Data Availability

The raw data supporting the conclusions of this article will be made available by the authors, without undue reservation.

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
