# Peer review of "Image- and Fluorescence-Based Test Shows Oxidant-Dependent Damages in Red Blood Cells and Enables Screening of Potential Protective Molecules"

_ijms, 2021, doi:10.3390/ijms22084293_

Round 1

Reviewer 1 Report

This manuscript describes a new analytical method called “TSOX” (test of sensitivity against oxidative stress) to analyze the effect of oxidizing stress, or the protective effect of antioxidant agents, against red blood cells (RBC). Several experimental approaches already exist to quantify oxidative stress of cell populations based on morphological analysis (microscopy or flow cytometry), metabolic, electrochemical analysis or ROS fluorescent probes such as 2'-7'-dichlorofluorescin diacetate (DCFH-DA), but a rapid and quantitative assay would be highly desirable to monitor the oxidative state of RBCs samples or to screen chemical libraries searching for compounds that confer protection from oxidative stress.

The method described in this study is composed of two assays: i) measure of the signal intensity of the fluorescent redox probe DCFH-DA, activated when intracellular ROS are generated; ii) morphology analysis by digital holographic microscopy (DHM), which provides two readouts: optical path difference (OPD) and confluency.

The aim of this work is to define a new bioanalytical method which, as such, would require an adequate experimental setup and validation plan. In the opinion of this referee, this work unfortunately presents some aspects of weakness that substantially decrease the value of the method proposed:

  • The novelty of an approach that combines techniques already available to analyze the oxidative stress of cellular samples is questionable: the Authors should comment on the added value of this method as compared to the other assays for the same application;
  • An orthogonal cell assay (DCFH-DA fluorescence + morphological analysis) might in principle provide more complete information than the individual assays that compose it, therefore it is not clear why in the proof-of-concept experiments shown in Figures 2 and 3 the same compounds were tested with the two readouts at different concentrations and for different treatment times. It would have been appropriate to analyze the same samples in parallel with the two readouts to show the different sensitivity, dynamic range and to emphasize the complementarity of the two assays. Since the experimental setup of DCFH-DA fluorescence was conducted under experimental conditions different from those of the morphology analysis, no convincing data is provided to demonstrate the superiority of this approach as compared to the two individual assays and it is not even clear whether "TSOX" can really be defined as an assay;
  • Since the "TSOX" approach (or, rather, the assays that compose it) is proposed for screening libraries of chemical compounds, it is appropriate to demonstrate the robustness of the method with statistical analysis (i.e., Z-factor) measuring the inter- and intra-experiment variability of low and high controls, also considering that a slight increase in the fluorescence signal of the DCFH-DA probe is observed over time in untreated controls (Figures 2A and 3A);
  • The Authors must provide more experimental details on the DHM T1000 time-lapse analysis, in particular about the temporal scanning of the frames and on the image analysis procedure, which is completely missing. The optical path difference (OPD) and confluency are very poorly described and it is not clear what they consist of.

Minor points:

  • Figure 2B: the doses of H2O2 in the graph (0.001% and 0.0001%) must be swapped;
  • What is the nature of the diamond-shaped "Tirocytes" shown in Figure 5C?

Author Response

RESPONSE FROM THE AUTHORS TO THE REVIEWER 1 COMMENTS

First, we would like to thank the reviewers for their remarks and comments that were carefully considered during the revisions of the manuscript. They helped us to improve it. The answers to the reviewer’s comments are listed below.

General comment on the message delivered by the article

Following the remarks of some of the reviewers, we decided to modify somewhat the message delivered by the article. Hence, rather than talking about an assay, we preferred the term “test” that better describes, from our perspective, the nature of the TSOX. And, in addition, we focused the discussion on particular effects of oxidant and antioxidants on RBCs.

Moreover, we agree with the fact that the test is not yet ready for HTS and would require further validation (i.e. determination of the Z’-factor). A note was added in the discussion about this particular point (rows 578-580). However, we believe that the observations made (i.e. effect of the oxidants and antioxidants on the RBCs) and the method that were used could per se be of interest and useful to other scientists.

Comments and Suggestions for Authors

This manuscript describes a new analytical method called “TSOX” (test of sensitivity against oxidative stress) to analyze the effect of oxidizing stress, or the protective effect of antioxidant agents, against red blood cells (RBC). Several experimental approaches already exist to quantify oxidative stress of cell populations based on morphological analysis (microscopy or flow cytometry), metabolic, electrochemical analysis or ROS fluorescent probes such as 2'-7'-dichlorofluorescin diacetate (DCFH-DA), but a rapid and quantitative assay would be highly desirable to monitor the oxidative state of RBCs samples or to screen chemical libraries searching for compounds that confer protection from oxidative stress.

The method described in this study is composed of two assays: i) measure of the signal intensity of the fluorescent redox probe DCFH-DA, activated when intracellular ROS are generated; ii) morphology analysis by digital holographic microscopy (DHM), which provides two readouts: optical path difference (OPD) and confluency.

The aim of this work is to define a new bioanalytical method which, as such, would require an adequate experimental setup and validation plan. In the opinion of this referee, this work unfortunately presents some aspects of weakness that substantially decrease the value of the method proposed:

  • The novelty of an approach that combines techniques already available to analyze the oxidative stress of cellular samples is questionable: the Authors should comment on the added value of this method as compared to the other assays for the same application;
  • An orthogonal cell assay (DCFH-DA fluorescence + morphological analysis) might in principle provide more complete information than the individual assays that compose it, therefore it is not clear why in the proof-of-concept experiments shown in Figures 2 and 3 the same compounds were tested with the two readouts at different concentrations and for different treatment times. It would have been appropriate to analyze the same samples in parallel with the two readouts to show the different sensitivity, dynamic range and to emphasize the complementarity of the two assays. Since the experimental setup of DCFH-DA fluorescence was conducted under experimental conditions different from those of the morphology analysis, no convincing data is provided to demonstrate the superiority of this approach as compared to the two individual assays and it is not even clear whether "TSOX" can really be defined as an assay;

We believe that the two methods are complementary. Indeed, we observed in some cases (such as with Resveratrol) that this antioxidant lowers the fluorescence generated under oxidative stress. However, looking at the morphology, we realized that reduction of the fluorescence signal was in fact attributable to cell lysis. In the future, we plan to combine the two readouts by using the fluorescent camera that can be mounted on the DHM. This particular point was discussed in more details in the discussion (rows 570-577).

  • Since the "TSOX" approach (or, rather, the assays that compose it) is proposed for screening libraries of chemical compounds, it is appropriate to demonstrate the robustness of the method with statistical analysis (i.e., Z-factor) measuring the inter- and intra-experiment variability of low and high controls, also considering that a slight increase in the fluorescence signal of the DCFH-DA probe is observed over time in untreated controls (Figures 2A and 3A);

Please refer to the “General comment on the message delivered by the article” above.

  • The Authors must provide more experimental details on the DHM T1000 time-lapse analysis, in particular about the temporal scanning of the frames and on the image analysis procedure, which is completely missing. The optical path difference (OPD) and confluency are very poorly described and it is not clear what they consist of.

More details were added both in the Results (section 2.1) and in the Material and Methods (section 4.3).

Minor points:

  • Figure 2B: the doses of H2O2 in the graph (0.001% and 0.0001%) must be swapped;

Figure 2B was corrected. Thanks for highlighting this mistake.

  • What is the nature of the diamond-shaped "Tirocytes" shown in Figure 5C?

We do not know the exact nature of those RBCs as it is the first time that we observed such kind of morphology. The discussion about the potential reasons leading to this particular shape change was extended (rows 521-543) after further review of the literature.

Reviewer 2 Report

The manuscript entitled “Image- and Fluorescence-based Assay to Test Red Blood Cells Sensitivity to Oxidative Stress and Screen Potential Protective Molecules” by Bardyn et. al. describe a development of a new method (TSOX) to study the effect of oxidants and antioxidants on RBCs. Authors have done a great job in developing this tool that would help screen for molecules having protective effects on RBC storage. This manuscript would benefit the field in further development of methods in accessing the quality and  storage conditions of transfusions.

Author Response

RESPONSE FROM THE AUTHORS TO THE REVIEWER 2 COMMENTS

First, we would like to thank the reviewers for their remarks and comments that were carefully considered during the revisions of the manuscript. They helped us to improve it. The answers to the reviewer’s comments are listed below.

General comment on the message delivered by the article

Following the remarks of some of the reviewers, we decided to modify somewhat the message delivered by the article. Hence, rather than talking about an assay, we preferred the term “test” that better describes, from our perspective, the nature of the TSOX. And, in addition, we focused the discussion on particular effects of oxidant and antioxidants on RBCs.

Moreover, we agree with the fact that the test is not yet ready for HTS and would require further validation (i.e. determination of the Z’-factor). A note was added in the discussion about this particular point (rows 578-580). However, we believe that the observations made (i.e. effect of the oxidants and antioxidants on the RBCs) and the method that were used could per se be of interest and useful to other scientists.

Reviewer 3 Report

The manuscript is decently written. Some words should be avoided in the scientific publications if there is no context for them, such as “progressive”, “diverse”, “ethnicity”. A phrase in the abstract is probably needed. It should indicate the importance of what is described in the manuscript. I have identified a few interesting observations (please see Very interesting notes of the manuscript that deserve a highlight) that should perhaps be inserted in the abstract; as I believe those are more important than the main subject of the manuscript. Also, the placing of the elements in figures is logical and pleasing to the reader. The presentation is professional and will be likely understood by a wide range of readers.

Minor observations:

  1. Please insert all references at the end of the phrase anywhere in the manuscript. It is annoying to see a reference in the middle of the phrase.
  2. Nevertheless, in the context of transfusion medicine, the storage of RBCs as RBC concentrates (RCCs) induces an oxidative stress that plays a major role in the apparition of the so called storage lesions” – I am of the opinion that this specific phrase (perhaps in a different form) should perhaps be the opening of the abstract. It shows the problem and the importance of the following information in the manuscript.
  3. In “Such disappointing results triggered the following reflection:” – Please add an “s” at the end of the word “reflection”. The authors show multiple points, so there is more than one reflection.
  4. Some from are very basic such as hematological parameters, hemolysis rate and …” – perhaps the word “from” should be changed to “forms”?
  5. other employee advanced technologies aiming” – please use “adopt” instead of “employee”. Also, perhaps other” should be “others”. Like for instance: “others adopt advanced technologies aiming”
  6. New ones like microfluidic devices …” – please change that to “New approaches like microfluidic devices …”.
  7. Here, we have designed a novel assay called TSOX (test of sensitivity against oxidative stress) able to determine if a compound provides protection to RBCs” – can you please add a “which is”? It will be like: “Here, we have designed a novel assay called TSOX (test of sensitivity against oxidative stress) which is able to determine if a compound provides protection to RBCs …”
  8. and at which concentration with a high-throughput capacity” – can you please rephrase that? It is not clear or the entire phrase is too long.
  9. Can you replace “It is based” with “TSOX is based” in “It is based on reactive oxygen species (ROS) detection …”?
  10. Note that you have two “based” words in the same phrase and that does not sound ok. “It is based on reactive oxygen species (ROS) detection by fluorescence and based on cell morphology by DHM.”. Please replace the second “based on” with “uses the”, or you can simply rephrase.
  11. Its flexible design can …” – please replace “Its” with “The”.
  12. For the TSOX, the average of the OPD distribution (OPD AVG) and confluency parameters were retained.” – where were these retained? please replace the word “retained” with “considered”, or rephrase.
  13. “Interestingly, some of the RBCs transformed into diamond-shaped “Tirocytes”” – this points out that crystal-like structures may form inside the cells due to resveratrol high concentrations. This is interesting and it shows the bad side of resveratrol. Perhaps this must be speculated a little further.
  14. triggered a progressive increase of fluorescence signal” – oh, the word progressive is annoying. This can be shortened to “triggered an increase of the fluorescence signal”.
  15. With a view to using the TSOX for screening purposes” – please rephrase that. Is not clear.
  16. The TSOX is a valuable tool to investigate both the aging of RBCs under various conditions, including the donors’ characteristics where differences are known in function of the sex or the ethnicity [33], [34], and discover hit molecules that could be added in blood bags to help better preserve RBCs for transfusion.” – by all means, please split this in two or three phrases. Also I can’t see the point of the word “ethnicity” since the whole paper presents a method and does not refer to particularities as such. Also it is advisable to avoid references in the text of the conclusion if possible.
  17. “diverse assay” – please delete the word “diverse”. It is annoying and incorrectly used here. The “a combination of assays” is clear and sufficient.
  18. Again “This tool also enables to further investigate the relation between metabolic pathways and cell morphology and combination of diverse assay design helps to understand the mode of action of toxic or protective compounds.” – the entire phrase should be split in two separate phrases for more clarity.
  19. AAPH treatment also ultimately leaded to cell lysis” – does it? “Indeed, cell lysis reduces fluorescence, as it was …” – this is visible below. I have taken the liberty to amplify the background of your panels from figure 2, to see if the fluorescence is lost/diluted in to the liquid do to cell lysis.

(Please see the image in the word document attached to this text)

The second row (Diamide 2mM) perhaps shows that cell lysis is present. The first line (AAPH) shows that the background contains no impurities regarding to the fluorescence liking out. Thus, there is no lysis involved or it is only partial. However, on the images on the second row (Diamide) you can see that the background remains somewhat constant. Thus, the Diamide lot shows cell lysis whereas, from the images, AAPH does not (apparently). How do you explain this?

Very interesting notes of the manuscript that deserve a highlight:

  1. “It is interesting to notice that fluorescence slightly increased in the control condition where no oxidant was added in the well.”
  2. “Despite the fact that H2O2 increased ROS generation within RBCs, its effect on morphology and more particularly on confluency was limited, even with higher concentrations”
  3. Interestingly, some of the RBCs transformed into diamond-shaped “Tirocytes” (this observation being more important than the manuscript itself)

Additional observation:

Regarding the “a quantitative value (i.e. optical path difference [OPD]) measured on each pixel of the phase image” – here I wish to point out to the authors a new method that can be used in these conditions, which is called “Spectral forecast” (https://aip.scitation.org/doi/10.1063/1.5120818 ), which can be used for different predictions on the behavior of the experiment in subsequent phases.

My conclusion on the whole manuscript:

In the context of transfusions, the storage of erythrocytes can be improved by adding resveratrol in the appropriate concentrations to avoid storage lesions. Higher concentrations of resveratrol may result in the formation of rigid cristal-like structures inside the erythrocyte cells.

Author Response

RESPONSE FROM THE AUTHORS TO THE REVIEWER 3 COMMENTS

First, we would like to thank the reviewers for their remarks and comments that were carefully considered during the revisions of the manuscript. They helped us to improve it. The answers to the reviewer’s comments are listed below.

General comment on the message delivered by the article

Following the remarks of some of the reviewers, we decided to modify somewhat the message delivered by the article. Hence, rather than talking about an assay, we preferred the term “test” that better describes, from our perspective, the nature of the TSOX. And, in addition, we focused the discussion on particular effects of oxidant and antioxidants on RBCs.

Moreover, we agree with the fact that the test is not yet ready for HTS and would require further validation (i.e. determination of the Z’-factor). A note was added in the discussion about this particular point (rows 578-580). However, we believe that the observations made (i.e. effect of the oxidants and antioxidants on the RBCs) and the method that were used could per se be of interest and useful to other scientists.

Comments and Suggestions for Authors

The manuscript is decently written. Some words should be avoided in the scientific publications if there is no context for them, such as “progressive”, “diverse”, “ethnicity”.

“Progressive” was replaced by “gradual”, “diverse” by “various”, and the sentence containing the reference to the donor’s ethnicity (conclusion) was removed.

A phrase in the abstract is probably needed. It should indicate the importance of what is described in the manuscript. I have identified a few interesting observations (please see Very interesting notes of the manuscript that deserve a highlight) that should perhaps be inserted in the abstract; as I believe those are more important than the main subject of the manuscript. Also, the placing of the elements in figures is logical and pleasing to the reader. The presentation is professional and will be likely understood by a wide range of readers.

Minor observations:

1. Please insert all references at the end of the phrase anywhere in the manuscript. It is annoying to see a reference in the middle of the phrase. All references were moved to the end of the sentences.

2. “Nevertheless, in the context of transfusion medicine, the storage of RBCs as RBC concentrates (RCCs) induces an oxidative stress that plays a major role in the apparition of the so called storage lesions” – I am of the opinion that this specific phrase (perhaps in a different form) should perhaps be the opening of the abstract. It shows the problem and the importance of the following information in the manuscript. The major role of oxidative stress was mentioned as proposed in the abstract (rows 20-21).

3. In “Such disappointing results triggered the following reflection:” – Please add an “s” at the end of the word “reflection”. The authors show multiple points, so there is more than one reflection. Done

4. “Some from are very basic such as hematological parameters, hemolysis rate and …” – perhaps the word “from” should be changed to “forms”? The word “from” was removed

5. “other employee advanced technologies aiming” – please use “adopt” instead of “employee”. Also, perhaps other” should be “others”. Like for instance: “others adopt advanced technologies aiming” The sentence was modified

6. “New ones like microfluidic devices …” – please change that to “New approaches like microfluidic devices …”. “Ones” was replaced by “approaches”, as proposed

7. “Here, we have designed a novel assay called TSOX (test of sensitivity against oxidative stress) able to determine if a compound provides protection to RBCs” – can you please add a “which is”? It will be like: “Here, we have designed a novel assay called TSOX (test of sensitivity against oxidative stress) which is able to determine if a compound provides protection to RBCs …”

8. “and at which concentration with a high-throughput capacity” – can you please rephrase that? It is not clear or the entire phrase is too long.

9. Can you replace “It is based” with “TSOX is based” in “It is based on reactive oxygen species (ROS) detection …”?

10. Note that you have two “based” words in the same phrase and that does not sound ok. “It is based on reactive oxygen species (ROS) detection by fluorescence and based on cell morphology by DHM.”. Please replace the second “based on” with “uses the”, or you can simply rephrase.

The last part of the introduction was rewritten and the corrections that you suggested were made.

11. “Its flexible design can …” – please replace “Its” with “The”. The sentence was modified

12. “For the TSOX, the average of the OPD distribution (OPD AVG) and confluency parameters were retained.” – where were these retained? please replace the word “retained” with “considered”, or rephrase. “retained” was replaced by “considered”, as proposed

13. “Interestingly, some of the RBCs transformed into diamond-shaped “Tirocytes”” – this points out that crystal-like structures may form inside the cells due to resveratrol high concentrations. This is interesting and it shows the bad side of resveratrol. Perhaps this must be speculated a little further. The discussion about the potential reasons leading to this particular shape change was extended (rows 521-543) after further review of the literature.

14. “triggered a progressive increase of fluorescence signal” – oh, the word progressive is annoying. This can be shortened to “triggered an increase of the fluorescence signal”. Modification was made

 15. With a view to using the TSOX for screening purposes” – please rephrase that. Is not clear. The sentence was rephrased (rows 549-551)

16. “The TSOX is a valuable tool to investigate both the aging of RBCs under various conditions, including the donors’ characteristics where differences are known in function of the sex or the ethnicity [33], [34], and discover hit molecules that could be added in blood bags to help better preserve RBCs for transfusion.” – by all means, please split this in two or three phrases. Also I can’t see the point of the word “ethnicity” since the whole paper presents a method and does not refer to particularities as such. Also it is advisable to avoid references in the text of the conclusion if possible.

Mention of ethnicity and references were removed from the conclusion.

 17. “diverse assay” – please delete the word “diverse”. It is annoying and incorrectly used here. The “a combination of assays” is clear and sufficient.

18. Again “This tool also enables to further investigate the relation between metabolic pathways and cell morphology and combination of diverse assay design helps to understand the mode of action of toxic or protective compounds.” – the entire phrase should be split in two separate phrases for more clarity.

The sentence was split in two and modified as requested

19. “AAPH treatment also ultimately leaded to cell lysis” – does it? “Indeed, cell lysis reduces fluorescence, as it was …” – this is visible below. I have taken the liberty to amplify the background of your panels from figure 2, to see if the fluorescence is lost/diluted in to the liquid do to cell lysis.

(Please see the image in the word document attached to this text)

The second row (Diamide 2mM) perhaps shows that cell lysis is present. The first line (AAPH) shows that the background contains no impurities regarding to the fluorescence liking out. Thus, there is no lysis involved or it is only partial. However, on the images on the second row (Diamide) you can see that the background remains somewhat constant. Thus, the Diamide lot shows cell lysis whereas, from the images, AAPH does not (apparently). How do you explain this?

In fact, the RBCs presented in the images were not treated with DCFH-DA (two separate readouts). The increase of background signal could rather due to the increase of hemoglobin in the supernatant because of cell lysis. However, with the low cell density used in our experiments (80’000 RBCs / 100 µL), we believe that the impact of lysis on background signal would probably be too low to be quantified. The question would deserve to be invested as we generally never measure the value of the background. Indeed, in the image analysis pipeline, a threshold is applied so that the OPD value is only measured on pixels occupied by cells. However, what is visible in case of cell lysis are the RBC ghosts that generally remain slightly visible by increasing the image contrast.

Very interesting notes of the manuscript that deserve a highlight:

  1. “It is interesting to notice that fluorescence slightly increased in the control condition where no oxidant was added in the well.”
  2. “Despite the fact that H2O2 increased ROS generation within RBCs, its effect on morphology and more particularly on confluency was limited, even with higher concentrations”
  3. Interestingly, some of the RBCs transformed into diamond-shaped “Tirocytes” (this observation being more important than the manuscript itself)

Lot of modifications were brought to the manuscript in order to better describe the results obtained with our test. We tried to keep in mind those notes during the rewrite. We hope that the modifications will highlight those points.

Additional observation:

Regarding the “a quantitative value (i.e. optical path difference [OPD]) measured on each pixel of the phase image” – here I wish to point out to the authors a new method that can be used in these conditions, which is called “Spectral forecast” (https://aip.scitation.org/doi/10.1063/1.5120818 ), which can be used for different predictions on the behavior of the experiment in subsequent phases.

My conclusion on the whole manuscript:

In the context of transfusions, the storage of erythrocytes can be improved by adding resveratrol in the appropriate concentrations to avoid storage lesions. Higher concentrations of resveratrol may result in the formation of rigid cristal-like structures inside the erythrocyte cells.

Your suggestion is very interesting, we added it in the discussion (rows 525-526).

Reviewer 4 Report

Major concern.

In the introduction (line 71) the authors state that they designed a “novel” assay, called TSOX. In my opinion the Authors only combines two methods already described in literature.

ROS evaluation by the fluorescent probe DCFH-DA is amply reported, using a single reading fluorimeter. Also the use of multiwell plates for the reading is already been reported (ref 15).

As far as DHM is concerned, Jiaqi Liu’ group explored the “Microdeformation of RBCs under oxidative stress measured by digital holographic microscopy and optical tweezers”. Appl Opt. 2019 20;58(15):4042-4046. Please add this paper in the reference list

The Authors should better discuss the additional information deriving from combining the two methods.

Minor concerns.

 From the M M paragraph (lines 264-266) it appears clear that Authors utilize stored RBC (42 days at 4°C) and this is not indicated in the abstract. In my opinion, freshly prepared RBC should be considered in this study.

All the experiments were performed by overnight treatment of RBC. In my opinion, based on the different mechanism of ROS production (direct or following thiols depletion), the time-dependence of the process should be investigated, particularly related to shorter incubation points.  In this respect, the authors report that fluorescence slightly increased in controls, indicating that overnight incubation  trigger spontaneous generation of endogenous ROS (line 116-118) according to reference finding that  RBC overnight incubation results in a significant increase in ROS produced.

The oxidant and antioxidant concentrations are quite high compared with the active oxidant/antioxidant concentrations reported for the selected compounds in literature. Please comment on which bases you choosed the selected concentrations.

In conclusion, for all these reasons, the “proof of concept” of the assay developed in this study aiming of assessing RBC oxidative damage and protection of antioxidants under various experimental conditions does not appear completely clear to me.

Author Response

RESPONSE FROM THE AUTHORS TO THE REVIEWER 4 COMMENTS 

First, we would like to thank the reviewers for their remarks and comments that were carefully considered during the revisions of the manuscript. They helped us to improve it. The answers to the reviewer’s comments are listed below.

General comment on the message delivered by the article

Following the remarks of some of the reviewers, we decided to modify somewhat the message delivered by the article. Hence, rather than talking about an assay, we preferred the term “test” that better describes, from our perspective, the nature of the TSOX. And, in addition, we focused the discussion on particular effects of oxidant and antioxidants on RBCs.

Moreover, we agree with the fact that the test is not yet ready for HTS and would require further validation (i.e. determination of the Z’-factor). A note was added in the discussion about this particular point (rows 578-580). However, we believe that the observations made (i.e. effect of the oxidants and antioxidants on the RBCs) and the method that were used could per se be of interest and useful to other scientists.

Major concern.

In the introduction (line 71) the authors state that they designed a “novel” assay, called TSOX. In my opinion the Authors only combines two methods already described in literature.

ROS evaluation by the fluorescent probe DCFH-DA is amply reported, using a single reading fluorimeter. Also the use of multiwell plates for the reading is already been reported (ref 15).

As far as DHM is concerned, Jiaqi Liu’ group explored the “Microdeformation of RBCs under oxidative stress measured by digital holographic microscopy and optical tweezers”. Appl Opt. 2019 20;58(15):4042-4046. Please add this paper in the reference list

The proposed reference was added in the reference list (ref [13]).

The Authors should better discuss the additional information deriving from combining the two methods.

We believe that the two methods are complementary. Indeed, we observed in some cases (such as with Resveratrol) that this antioxidant lowers the fluorescence generated under oxidative stress. However, looking at the morphology, we realized that reduction of the fluorescence signal was in fact attributable to cell lysis. In the future, we plan to combine the two readouts by using the fluorescent camera that can be mounted on the DHM. This particular point was discussed in more details in the discussion (rows 570-577).

Minor concerns.

From the M M paragraph (lines 264-266) it appears clear that Authors utilize stored RBC (42 days at 4°C) and this is not indicated in the abstract. In my opinion, freshly prepared RBC should be considered in this study.

In fact, we used RCCs stored for less than 12 days. Clarification about the type of products used in this study was made in the material and methods (rows 571-572).

All the experiments were performed by overnight treatment of RBC. In my opinion, based on the different mechanism of ROS production (direct or following thiols depletion), the time-dependence of the process should be investigated, particularly related to shorter incubation points. In this respect, the authors report that fluorescence slightly increased in controls, indicating that overnight incubation trigger spontaneous generation of endogenous ROS (line 116-118) according to reference finding that RBC overnight incubation results in a significant increase in ROS produced.

It is true. We further discussed this point in the discussion (rows 468-479).

The oxidant and antioxidant concentrations are quite high compared with the active oxidant/antioxidant concentrations reported for the selected compounds in literature. Please comment on which bases you choosed the selected concentrations.

The concentrations used were selected on the basis of the numerous preliminary experiments we made before the “miniscreen” presented in this article.

In conclusion, for all these reasons, the “proof of concept” of the assay developed in this study aiming of assessing RBC oxidative damage and protection of antioxidants under various experimental conditions does not appear completely clear to me.

Round 2

Reviewer 1 Report

The authors have satisfactorily addressed the points I raised in the first review, and in particular they have placed less emphasis on the claim of having developed a cell assay suitable for compound screening, for which however they have not provided any convincing validation data.

Consideration for publication of the manuscript in this form is recommended.

Reviewer 4 Report

I have carefully reviewed the revised version of the manuscript and I think that the Authors significantly improved the manuscript that is suitable for publication in the present modified form.